# Resolving Sahelian thunderstorms improves mid-latitude weather forecasts

Gregor Pante [1] & Peter Knippertz [1]

The summertime West African Sahel has the worldwide highest degree of thunderstorm organisation into long-lived, several hundred-kilometre elongated, fast propagating systems that contribute 90% to the annual rainfall. All current global weather prediction and climate models represent thunderstorms using simplified parameterisation schemes which deteriorates the modelled distribution of rainfall from individual storms and the entire West African monsoon circulation. It is unclear how this misrepresentation of Sahelian convection affects forecasts globally. Our study is the first to demonstrate how a computationally feasible increase of model resolution over West Africa – allowing to avoid convection parameterisation – yields a better representation of organised convection in the Sahel and of moisture within the monsoon system, ultimately improving 5–8-day tropical and mid-latitude weather forecasts. We advocate an operational use of a modelling strategy similar to the one presented here for a cost-effective improvement of global weather prediction and potentially even (sub-)seasonal and climate simulations.

[1] Institute of Meteorology and Climate Research, Karlsruhe Institute of Technology, Karlsruhe, Germany. Correspondence and requests for materials should be addressed to G.P. (email: gregor.pante@kit.edu)

Observed teleconnections between the West African monsoon (WAM) region and the North Atlantic/European sector[1,2] and known shortcomings in representing Sahelian thunderstorms with convection parameterisations[3–5] point to a potentially positive effect of explicitly resolving convection in the Sahel on weather forecasts over Europe.

Thunderstorms, or more precisely mesoscale convective systems (MCSs), mostly of squall-line type[6], control Sahelian rainfall[7]. By definition MCSs have lifetimes of several hours to a few days, horizontally extend over more than 5000 km$^2$ and propagate at more than 10 m s$^{-1}$ [8]. The misrepresentation of MCSs by convection parameterisations in numerical weather prediction (NWP) and climate models[4,9] has negative implications for the simulated diurnal cycle and location of precipitation and, consequently, soil moisture and atmospheric circulation in the WAM region[3,10]. One past attempt to overcome systematic failures in NWP over West Africa was to improve initial conditions in the Sahel using additional observations from the African Monsoon Multidisciplinary Analysis field campaign during the 2006 wet monsoon season[11]. Indeed, a forecast improvement over Europe after some days lead time could be revealed[12].

In this study, we use the seamless NWP and climate ICOsahedral Nonhydrostatic (ICON) modelling framework[13,14]. Using a grid spacing of 13 km—as in the operational NWP model set-up at the German Weather Service (DWD)—we perform two different types of simulations: the first uses the Tiedtke–Bechtold convection parameterisation[15,16] for the whole globe, hereafter abbreviated with PARAM. The other employs a two-way nesting domain with a refined grid of 6.5 km over West Africa (4°S–28°N, 22°W–28°E; red box in Supplementary Fig. 1), where the parameterisations for deep and shallow convection are turned off (referred to as EXPLC). Two-way nesting means an exchange of information between different model domains through relaxation. For both PARAM and EXPLC we performed in total 124 10-day forecasts initialised every day in July and August 2016 and 2017 at 12 UTC (Coordinated Universal Time). For July 2016 additional experiments using a range of model set-ups (Supplementary Table 1) reveal the sensitivity of the results to model version and initialisation (Supplementary Note 1).

Comparing EXPLC with PARAM simulations shows that explicit convection leads to improved forecasts in the Sahel and that teleconnections to forecasts in the wider tropics and extratropics exist. Improvements over Africa imply improved forecasts over adjacent extratropical regions, while impacts on the rest of the tropics are more mixed. We conclude that expanding the sparse observational network in West Africa and targeted model developments to improve the representation of Sahelian thunderstorms will be beneficial for global weather and potentially even longer-term predictions.

## Results

**Impact of Sahelian MCSs on forecasts for West Africa.** Our first objective is to analyse if the two-way nesting leads to systematic improvements of forecasts over West Africa itself. Since analysis data produced with a global model using convection parameterisation may suffer from unrealistic WAM dynamics[17], the evaluation will be done exclusively with observations.

PARAM generates widespread light rainfall across the Sahel and only weak and slowly moving systems (Fig. 1a). Allowing for explicit convection, the model simulates several individual, intense MCSs crossing the Sahel westwards with a lifetime of 2–4 days (Fig. 1b), which compares well to Tropical Rainfall Measuring Mission (TRMM 3B42[18,19]) observations (Fig. 1c) in terms of timing, intensity and propagation speed. An exact match for individual systems, however, cannot be expected from a 10-

day forecast. This result holds even for explicit convection at 13 km grid spacing (see Supplementary Fig. 2). EXPLC reduces the large wet bias over the Sahel seen in PARAM and also much better reproduces the observed rainfall peak in the late afternoon to early evening (Fig. 1d). Despite this consistent improvement over the Sahel, EXPLC also enhances existing dry biases to the south of the main rain band, particularly to the west of the Guinea Highlands and over the Niger Delta region (Fig. 1e, f). As convection is more locally triggered by topographic features and therefore less organised there, we speculate that the employed grid spacing of 6.5 km may not be adequate, calling for experiments with higher resolution. In fact, recent work has shown substantial model biases even with 4.5 km grid spacing and large observational uncertainties in these regions[20]. Largely following the above-mentioned patterns in rainfall changes, EXPLC improves forecasts of top-of-atmosphere infrared brightness temperatures to the north of 12°N and deteriorates them farther south (Supplementary Fig. 3c). However, due to the much better diurnal cycle and convective organisation, forecast errors after removing the relatively large biases (Supplementary Fig. 3a, b) are generally smaller in EXPLC (Supplementary Fig. 3d). Such mixed effects of partly resolved convection have been observed for other models, too[21].

In addition, the forecasts are compared with the four radiosonde stations in the region with data for at least 40% of the days in July and August 2016 and 2017 (Abidjan, Ouagadougou, Niamey, Tamanrasset; see Fig. 1 for locations). For forecast days 2–5, a substantial moist bias exists in PARAM for all stations and levels, except for the 250–600 hPa layer over Ouagadougou and the mid-troposphere over Tamanrasset (Fig. 2). The vertical profile of Tamanrasset may indicate problems in the model to transport low-level moisture out of the deep Saharan boundary layer into the free troposphere above, for example, by overshooting plumes[22]. Due to its high elevation of 1385 m, however, the representativity of this station for the wider Sahara is unclear. Moving to explicit convection significantly reduces the moist bias at Abidjan and Niamey through most of the troposphere, which should also positively impact cloud cover. Separate analysis for specific humidity and temperature reveals that this result is largely due to a reduction in column moisture compensated partly by cooler temperatures (Supplementary Fig. 4). This result is consistent with more idealised work demonstrating overall drying, when convection becomes more organised[23–25]. This drying trend continues during forecast days 6–10, causing a dry bias in EXPLC at Abidjan and Ouagadougou between 250 and 600 hPa, while temperature and wind signals do not change much anymore (Supplementary Figs. 5 and 6). In stark contrast to that, the low-level moisture increase (both relative and absolute; Fig. 2d, Supplementary Fig. 4j) in EXPLC at Tamanrasset may be an indication of the transport of cool and moist air into the Sahara by convective cold pools[3], possibly exacerbating biases resulting from boundary-layer problems.

The sparse observational network in West Africa hampers a more detailed evaluation of the forecasts in this region. Nevertheless, the comparison with satellite data and radiosondes indicates overall more realistic results in EXPLC over the Sahel with a somewhat more mixed performance over southern West Africa. It should be noted that in the two-way nesting domain the convection parameterisation is switched off, but the remaining model configuration corresponds to the operational set-up tuned to produce optimal results with the parameterisation. A re-tuning might, for instance, counteract the too strong drying in EXPLC as visible at Abidjan and Ouagadougou (Supplementary Fig. 5a, b). An interesting implication of this is that a configuration optimised for a grid spacing of 6.5 km and initialised from an analysis produced

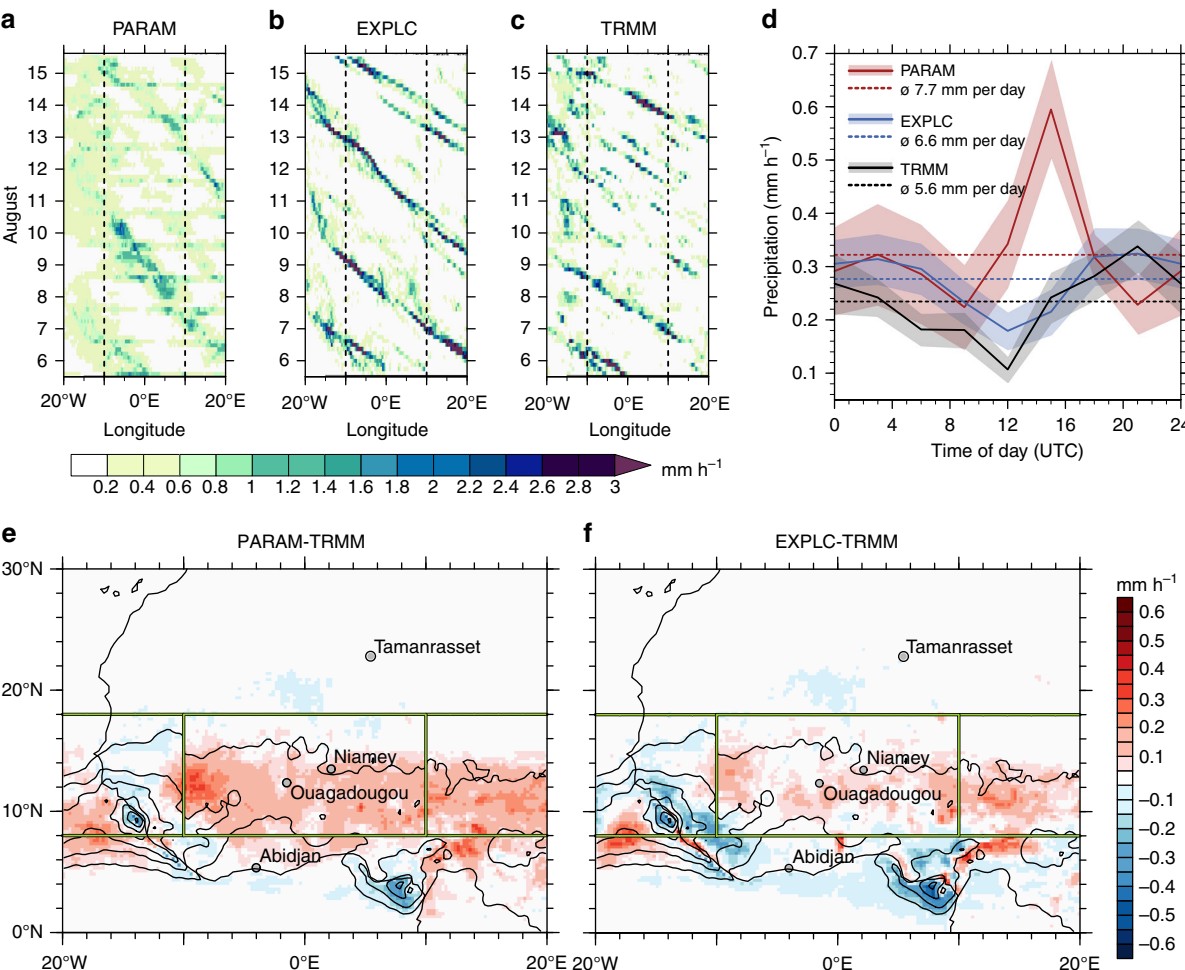

**Fig. 1** Model performance in simulating precipitation in West Africa. **a–c** Hovmoeller diagrams of 3-hourly precipitation averaged from 8°N to 18°N for the time period from 12 UTC (Coordinated Universal Time) 5 August 2017 (initialisation of ICOsahedral Nonhydrostatic (ICON simulations) to 12 UTC 15 August 2017 for model simulations employing the convection parameterisation (PARAM, **a**) and explicitly resolving convection (EXPLC, **b**), as well as Tropical Rainfall Measuring Mission (TRMM) observations (**c**). **d** Diurnal cycle of precipitation averaged from 8°N to 18°N and from 10°W to 10°E (dashed vertical lines in **a–c** and boxes in **e**, **f**). Shown are averages (bold lines) ± standard deviation (shading) over all 124 simulations of the 10-day forecast's mean diurnal cycle and the respective average for TRMM observations. **e** Precipitation bias for PARAM over all 124 simulations with averaging box from **d** and latitude band from **a** to **c** as well as radiosonde station locations used in this study indicated. Solid black lines show the corresponding TRMM climatology in steps of 0.2 mm h⁻¹. **f** As **e** but for EXPLC

with explicit convection over West Africa may further improve forecasts, but this remains to be proven.

**Remote impacts of Sahelian MCSs.** The interesting question now is if and how the improvement over West Africa affects the simulations outside of the two-way nesting domain. The difference of EXPLC minus PARAM with respect to 500-hPa geopotential height illustrates the spreading of the signals beyond West Africa for forecast days 5–8 (Fig. 3a). Absolute differences (contours) show a lowering of geopotential through large parts of the tropical belt, suggesting a cooler lower troposphere. This is associated with a substantial reduction of the root mean squared error (RMSE) (shading) in equatorial regions outside of Africa. The largest absolute differences are found in the southern hemisphere (i.e. winter) extratropics with a marked increase over the southeastern Atlantic and a decrease over the southwestern Indian Ocean. The structure of this signal resembles a Rossby-ray emanating from the subtropical South Atlantic, roughly following a great circle[1,26]. The impact on the RMSE is not as clear-cut with improvements in the vicinity of southern Africa but also

deterioration elsewhere. This indicates that due to the chaotic nature of the mid-latitude atmosphere, 4 months of simulation may not be enough to fully establish a potential improvement against the background weather noise. Over the northern hemisphere, a similar wave train exists with reduced geopotential over the subtropics followed by a marked increase over the mid-latitude Atlantic and a reduction over northeastern Europe. Impacts on RMSE are again rather noisy with the largest improvement stretching from Scandinavia to Russia.

To further investigate the statistical significance of these changes, Fig. 3b–d shows the relative improvement of the RMSE in EXPLC compared to PARAM (see figure caption for definition) averaged over all 124 simulations and the three orange bordered regions in Fig. 3a. The evaluation domains were chosen to represent the regions that are influenced most by the two-way nesting as discussed above. The domain in the southern hemispheric extratropics extends farther east than the European domain because signals propagate faster within the wintertime storm track. The distance to the West African domain avoids possible artefacts close to the two-way nesting boundary. The largest differences between EXPLC and PARAM are found for

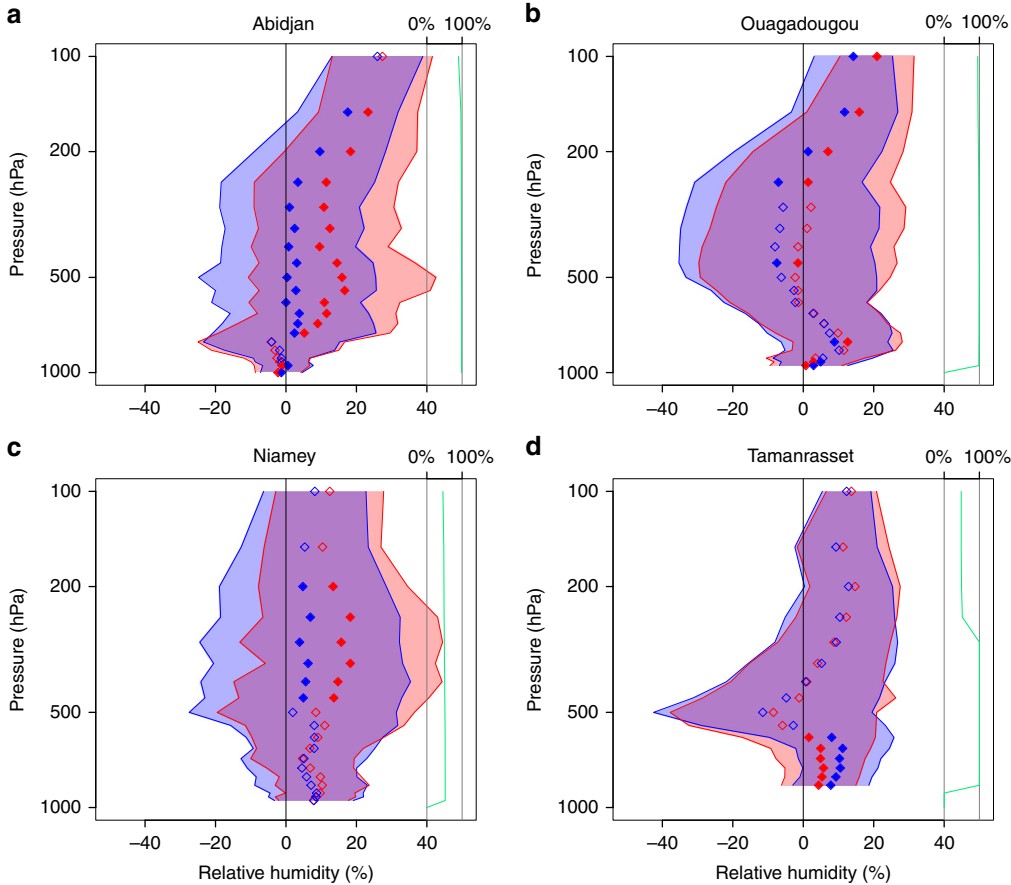

**Fig. 2** Model evaluation against radiosondes for relative humidity. Bias of relative humidity compared to radiosonde data from Abidjan (**a**), Ouagadougou (**b**), Niamey (**c**) and Tamanrasset (**d**). Averaged over all 124 simulations, shaded areas show the mean bias ± standard deviation of model simulations employing the convection parameterisation (PARAM, red) and explicitly resolving convection (EXPLC, blue) for all available soundings during forecast days 2–5. The mean values of the bias are marked by red and blue diamonds, which are filled if the difference between EXPLC and PARAM is statistically significant on the $\alpha = 5\%$ level and open otherwise. The scale to the right shows data availability in percent, that is, how many of the 124 days have at least one sounding a day at 00 or 12 UTC (Coordinated Universal Time)

forecast days 5–8 (shown in Fig. 3b–d). Before that, time is likely too short for signals to propagate from West Africa to the mid-latitudes and around the tropical belt. After day 8, forecasts slowly converge towards climatological values. In the European and South Atlantic/Indian Ocean domains, a mean improvement of the RMSE of up to 1% is visible in geopotential, temperature, specific humidity, wind speed and vector throughout the troposphere (Fig. 3b, c). A paired $t$-test reveals that the improvements are statistically significant at the 5 or 20% level for many of the variables and pressure levels shown. In the lower stratosphere at 100 hPa the effects are somewhat larger over Europe, possibly due to a higher tropopause in the summer hemisphere. In comparison, the improvement of a mean measure of 7-day forecast skill for the entire northern hemisphere at the European Centre for Medium-Range Weather Forecasts (ECMWF) increased from 45 to 75% in the 34-year period from 1981 to 2014[27]. Therefore, the improvement in the extratropics shown here points to a considerable potential to improve medium-range weather forecasts in the mid-latitudes through explicitly resolving organised convection over Africa, which should be further explored.

Compared to the extratropics, the relative effects of the nesting are generally larger in the tropical belt (Fig. 3d). Strikingly geopotential at 850 hPa is much deteriorated, while higher levels show moderate to large improvements in EXPLC. This is consistent with moderately better low-level temperature forecasts.

However, at 500 and 250 hPa RMSEs in temperature increase with a clear deterioration near the tropopause at 100 hPa. This indicates some issues with the vertical structure of the tropical troposphere in the model. Moisture fields in contrast are improved at low levels and near the tropopause, while wind signals show the opposite behaviour. Overall, these results indicate a complex response of the tropical troposphere to explicit convection that needs further study.

**Teleconnection mechanism.** How are the changes over Africa communicated to remote regions? As an integral measure of atmospheric changes between EXPLC and PARAM, the vertically integrated difference total energy (DTE, see Methods) is used. A Hovmoeller diagram for 40–0°W (Fig. 4a), the area of the Rossby wave train connecting the low and high latitudes of the northern hemisphere (see Fig. 3a), reveals that during the first days of the simulations the main DTE signal coincides with the Sahelian rain band (8–18°N, marked by dashed lines). Differences intensify and mainly spread northwards throughout the simulation, suggesting effects on the Saharan heat low (SHL) and the Azores high (see Supplementary Figs. 1 and 7a, c). The tropical easterly jet (TEJ) to the south of 10°N is mainly affected at later stages. In the extratropics the largest impact during the second half of the simulations is near the polar jet, where perturbations can grow baroclinically. From here the signal spreads even further north

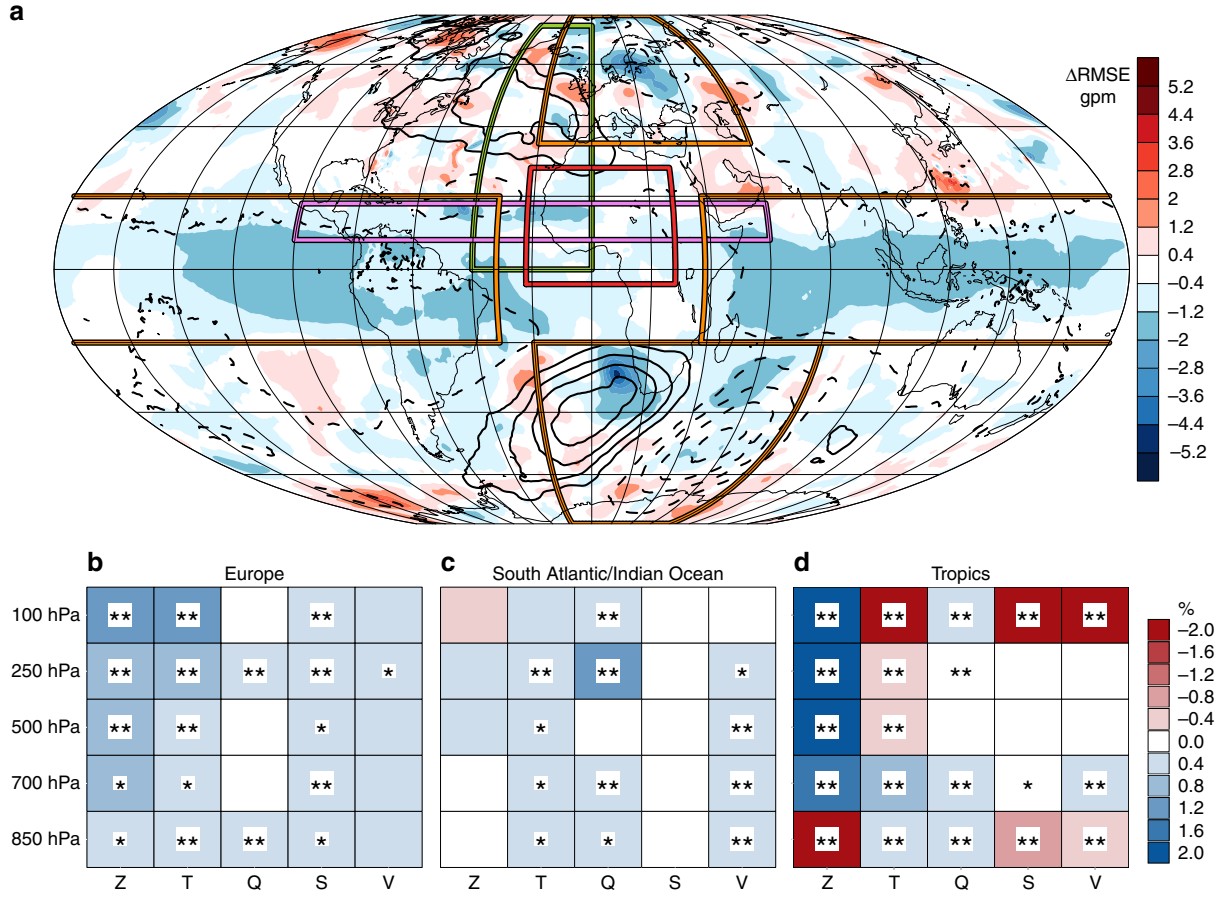

**Fig. 3** Impact of resolving Sahelian thunderstorms on global forecasts. **a** Difference of simulations explicitly resolving convection (EXPLC) minus simulations employing the convection parameterisation (PARAM) of geopotential height at 500 hPa averaged over forecast days 5–8. Differences of the root mean squared error (RMSE, shading) and absolute values (black contour lines in ±2 geopotential meter (gpm) intervals, negative dashed) are shown. The two-way nesting domain and the evaluation regions are given as red and orange boxes, respectively. Violet and green boxes mark averaging regions used to analyse signal propagation in zonal and meridional direction. **b**–**d** Relative improvement of the RMSE (i.e. (RMSE(EXPLC) − RMSE(PARAM))/RMSE(PARAM) ×100) in the European (**b**), South Atlantic/Indian Ocean (**c**) and tropical (**d**) evaluation regions, averaged over forecast days 5–8 for the variables geopotential (Z), temperature (T), specific humidity (Q), wind speed (S) and wind vector (V) at five different pressure levels. Double asterisks and single asterisks symbol denotes statistical significant differences between EXPLC and PARAM on the $\alpha = 5\%$ (20%) level. Blue denotes an improvement in EXPLC compared to PARAM, and red a deterioration. Time averaging considers the autocorrelation of the differences between EXPLC's and PARAM's RMSE (see Methods)

and to the ground (Supplementary Fig. 7c). Changes in strength and structure of the jet generally have a strong control on forecasts for Europe. Differences in the subtropical jet, which in contrast lies over the southeastern Mediterranean Sea at this time of the year (Supplementary Fig. 1), have only a limited influence on Europe.

Latitudinal averages of integrated DTE over the Sahelian rain band (Fig. 4b) reveal the immediate impact in the two-way nesting domain itself (22°W–28°E). Differences mostly propagate westward with a mean speed of 7° longitude per day (dashed blue line in Fig. 4b), typical of African easterly waves (AEWs)[28]. This confirms a hypothesis put forward by Faccani et al[12]. The vertical structure of DTE between 8°N and 18°N (Supplementary Fig. 7b, d) supports the idea of signal propagation along the African easterly jet (AEJ, 500–700 hPa) as well as close to the TEJ at upper levels. This is confirmed by clear DTE maxima immediately to the south of the TEJ and AEJ cores inside the two-way nesting domain (Fig. 4c). These jets act as wave guides (mainly AEJ) or cause a fast advection of signals generated over Africa. The faster spread of signals both westward and eastward may be connected to equatorial (e.g. Rossby and Kelvin) waves[29]. Once the area of upper-level westerlies over the Atlantic Ocean is reached, signals

can amplify in the free troposphere and induce a wave-like response in the atmosphere from the subtropical Atlantic to the polar jet region as visible in Fig. 3a.

In addition, a large DTE signal is found near the surface over the Sahara, demonstrating that explicit convection in the Sahel affects the SHL as discussed by Marsham et al[3]. Support for the importance of the SHL comes from comparing EXPLC- and PARAM-type simulations with different model set-ups (for July 2016 only). Improvements over Europe depend on the model version and initialisation data. The set-ups with the strongest impact on the SHL and, consequently, the AEJ, AEWs and the Azores high during the first half of the simulations show the most positive effect on the forecasts over Europe (Supplementary Note 1, Supplementary Fig. 8 and Supplementary Table 2).

**Changes over West Africa.** Finally, how does Sahelian convection influence the SHL, TEJ and AEJ that carry forecast improvements out of Africa? Previous work has shown impacts of explicit convection on the distribution and diurnal cycle of precipitation, the Sahel-Sahara pressure gradient and cold storm outflows[3]. To better understand the manifold interactions of various atmospheric features, we analyse differences between

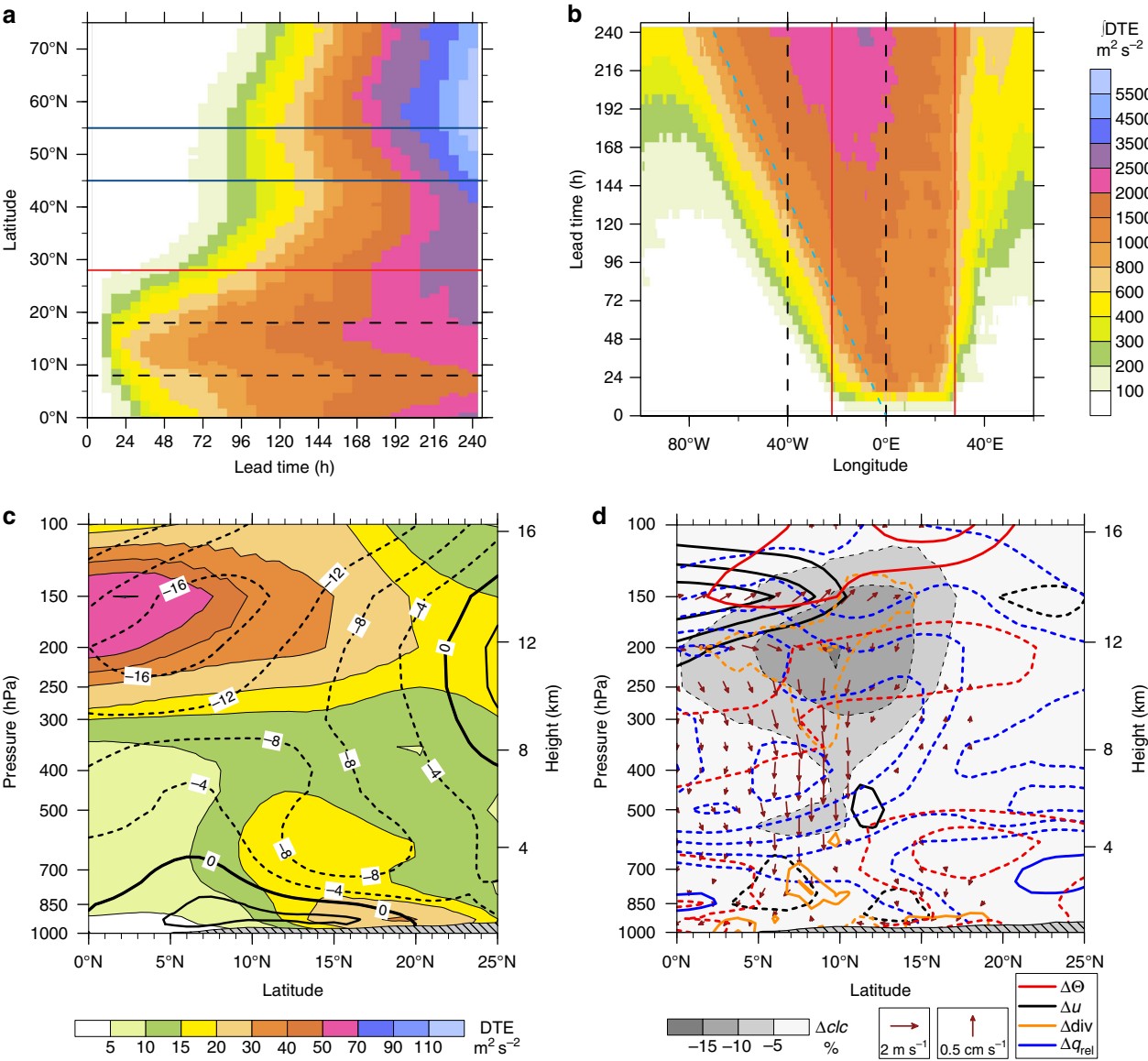

**Fig. 4** Mechanism connecting West Africa and remote regions. **a**, **b** Hovmoeller diagrams of the difference total energy (DTE) between simulations explicitly resolving convection (EXPLC) and simulations employing the convection parameterisation (PARAM) vertically integrated from 1000–100 hPa and averaged from 40°W to 0°W (**a**) and from 8°N to 18°N (**b**). Dashed black lines mark the respective other averaging regions, red lines the borders of the two-way nesting domain and the light blue dashed line in **b** illustrates the westward propagation of the differences. The polar jet maximum lies between the two blue solid lines in **a**. **c**, **d** Vertical cross-sections averaged from 10°W to 10°E and over all 10 forecast days with hatched grey shading showing the orography: **c** DTE (shading) and zonal wind speed of PARAM (contour lines labelled in m s$^{-1}$); **d** differences (EXPLC − PARAM) in cloud cover ($\Delta clc$, shading, no positive values are visible because maximum difference is +3%), meridional $\Delta v$ and vertical $\Delta w$ wind speed (vectors, only if $|\Delta v|$ >0.5 m s$^{-1}$ or $|\Delta w|$ >0.1 cm s$^{-1}$), and as contour lines (positive solid, negative dashed) potential temperature ($\Delta\Theta$, red: ±0.4 K intervals), zonal wind speed ($\Delta u$, black: ±1.5 m s$^{-1}$ intervals), divergence ($\Delta div$, orange: ±1.5 × 10$^{-6}$ s$^{-1}$ intervals) and the relative difference (i.e. (EXPLC − PARAM)/PARAM) of specific humidity ($\Delta q_{rel}$, blue: ±5% intervals)

EXPLC and PARAM in clouds, circulation, temperature and humidity over West Africa (Fig. 4d).

In the upper troposphere explicit convection reduces cloud cover by up to 16% (grey shading in Fig. 4d) and lowers the mean cloud top altitude (not shown). This causes a shift of the maximum cooling by outgoing longwave radiation to lower levels in EXPLC and consequently a vertical dipole of warmer and colder temperatures between 100 and 300 hPa. Recall from the earlier discussion that this implies a better agreement with satellite data over the Sahel, but a larger bias farther south (Supplementary Fig. 3). Weaker updraughts in the convectively active zone (about 8–12°N) are associated with less divergence

and outflow at upper levels. Due to the Coriolis effect the upper-level northerly outflow contributes to the TEJ, which in consequence is decelerated in EXPLC. The weakening of the TEJ of as much as 5 m s$^{-1}$ is directly associated with weaker temperature advection of cooler air from the east (not shown), which adds to the radiative effect above cloud top, expanding the warmer region to 2°N at 150 hPa.

The weaker upper-level divergence in EXPLC together with weaker convergence (i.e. a positive difference in divergence) at around 800 hPa influences the horizontal and vertical advection of moisture from the Gulf of Guinea. In combination with reduced convective vertical mixing, a drying of more than 25% in

the mid-troposphere results. The cooling south of 8°N centred around 350 hPa is likely the result of less longwave absorption and re-emission due to less clouds above and less absorption in the drier atmosphere below. The latter effect does not lead to a cooling in the entire lower and middle troposphere, because it is compensated by less ascent (corresponds to warming) and less precipitation (see Fig. 1d) causing less evaporative cooling below the clouds. The only region in this cross-section with more clouds in EXPLC lies south of 2°N at about 850 hPa (up to 3%, not visible due to ±5% contour interval). This increased cloudiness implies less incoming solar radiation explaining the cooling here.

Between 14°N and 18°N, EXPLC generates about 8% more rainfall than PARAM (not shown) despite a lower specific humidity and cloud cover. This indicates a much higher precipitation efficiency of MCSs in EXPLC with their more concentrated, stronger rainfall. In contrast, the convection parameterisation produces many individual clouds with moderate rain rates (see Fig. 1a–c) depositing clouds and moisture at middle and upper levels through detrainment. Stronger upward motion in EXPLC at 250–400 hPa and stronger downward motion below 700 hPa indicate up- and downdrafts of MCSs. Previous work[3,30–32] suggests that the convection parameterisation misses cold pool outflows from such storms resulting in an underestimation of the ventilation of the SHL in PARAM (see the discussion of the Tamanrasset radiosonde above). This effect—together with the reduction of specific humidity in EXPLC over the Sahara by 10–15% around 500 and 200 hPa, implying less absorption and downward emission of longwave radiation—cools the Saharan planetary boundary layer below 500 hPa. The SHL is therefore colder, that is, weaker, in EXPLC. Consequently, near-surface convergence is weaker between 15°N and 19°N and westerly winds south of the SHL are decelerated. A cooler Saharan boundary layer, in turn, reduces the strength of the AEJ, which forms as a thermal wind between the hot Sahara and the colder Sahel and controls characteristics of AEWs[33].

## Discussion

In this study, we demonstrate that explicitly resolving summertime Sahelian MCSs in the ICON model significantly impacts on forecast biases and errors over West Africa itself but also over remote regions in the tropics and extratropics, including socioeconomically important Europe. The novel two-way nesting approach allows us to realise these impacts in a quasi-operational setting for the first time with only a moderate increase in computation time of about 20%. Comparisons with observations show a much better representation of precipitation features and their diurnal cycle over the Sahel and an overall improvement of the moist bias in ICON, likely through an enhanced precipitation efficiency. However, problems over southern parts of West Africa and with the Saharan boundary layer remain or even deteriorate. Through a complicated combination of radiative and circulation mechanisms, these changes impact on the SHL, the AEJ and TEJ, which quickly carry signals out of Africa. The impacts are felt over the tropical belt and along Rossby wave trains arching into the extratropics of both hemispheres. Here the positive impacts are competing with the overall high background weather noise, but still lead to statistically significant mean reductions in 5–8-day RMSE of up to 1% over Europe and the southern Atlantic and Indian Oceans.

These results suggest several measures to further enhance the potential for forecast improvement unveiled here for the first time, namely, to create an analysis for West Africa based on convection permitting forecasts to hopefully cure the large moisture bias and to provide better initial conditions and products for model evaluation and to repeat the simulations analysed

here with a re-tuning of other model parameters, switching off the deep convection parameterisation only and with a grid spacing of only few kilometres over West Africa to potentially cure problems with more localised convection in southern West Africa. Furthermore, it would be interesting to extend the study to more years to reach better statistics in the extratropics and to investigate impacts on the global scale in subseasonal or even climate simulations, which also suffer from parameterised convection[34] or to apply our strategy to other hot spot regions for tropical thunderstorms such as the maritime continent. Operational weather prediction centres aim to globally resolve convection within the next decade. As an intermediate step, weather services are encouraged to evaluate the approach demonstrated here against more conventional strategies for forecast improvement.

## Methods

**ICON model set-up**. For the PARAM simulations, we use the operational set-up of the global ICON model with a two-way nesting domain over Europe (29.5–70.5°N, 23.5°W–62.5°E). The impact of explicit convection in the Sahel is investigated by only one change, namely, the incorporation of a second two-way nesting domain over West Africa (4°S–28°N, 22°W–28°E, red box in Supplementary Fig. 1), for which the parameterisations of deep and shallow convection are switched off (EXPLC simulations). All other namelist settings for global and nesting domains conform with the operational configuration. For the global (nesting) domain, this comprises a horizontal grid spacing of 13 km (6.5 km), a time step of 120 s (60 s) and 90 (60) vertical levels up to 75 km (22.5 km)[14]. Due to the additional nesting domain, EXPLC requires about 20% more CPU (central processing unit) time than PARAM.

The only modification of the source code is the incremental feedback of rain rates between the West African two-way nesting domain and the global domain for the EXPLC simulations. Rain rates are not directly coupled between model domains in official distributions where only the feedback of the prognostic variables (horizontal and vertical wind, virtual potential temperature, total air density, specific humidity, cloud water and cloud ice) is active. For the West African domain this feedback for precipitation in EXPLC simulations was introduced in order to not only affect atmospheric dynamics but also soil moisture, which directly influences the initiation of deep convection in this region[35].

For the global domain, various atmospheric variables are output six-hourly at 0, 6, 12 and 18 UTC on a regular 0.75° × 0.75° grid on different vertical pressure levels for all model set-ups. Precipitation is output hourly on a regular 0.25° × 0.25° grid for comparison with TRMM observations.

**Definition of error scores**. The systematic analysis of the impact of explicitly resolving convection in the Sahel requires the evaluation of the simulations using error scores. Standard scores for the verification of deterministic NWP products are the mean error (or bias) and the RMSE according to Eqs. (1) and (2), respectively. Following the World Meteorological Organization guideline, we calculate the RMSE for geopotential, temperature, moisture and wind variables on different vertical levels[36] for the evaluation regions as marked in Fig. 3a, namely, Europe (35–80°N, 20°W–60°E), South Atlantic/Indian Ocean (20–80°S, 20°W–80°E) and the tropics without the nesting domain (20°S–20°N, excluding 32°W–38°E). These borders prevent possible artefacts of the nesting close to the boundary zone to be part of the evaluation regions.

$$\text{Bias} = \frac{1}{n}\sum_{i=1}^{n} w_i\left(x_i^{\text{f}} - x_i^{\text{a}}\right), \qquad (1)$$

$$\text{RMSE} = \frac{1}{n}\sqrt{\sum_{i=1}^{n} w_i\left(x_i^{\text{f}} - x_i^{\text{a}}\right)^2}, \qquad (2)$$

$$\text{CRMSE} = \frac{1}{n}\sqrt{\sum_{i=1}^{n} w_i\left[\left(x_i^{\text{f}} - \overline{x^{\text{f}}}\right) - \left(x_i^{\text{a}} - \overline{x^{\text{a}}}\right)\right]^2}, \qquad (3)$$

where $x^{\text{f}}$ is the forecast value, $x^{\text{a}}$ the corresponding analysis value, $w_i$ the cosine of latitude at grid point $i$ and $n$ the number of grid points in the respective evaluation domain. A computation of the RMSE after removing the bias is referred to as centred RMSE (CRMSE), where $\overline{x^{\text{f}}}$ and $\overline{x^{\text{a}}}$ are the area-averaged values of the respective fields. All forecasts are evaluated against the analysis used for model initialisation, that is, initDWD against DWD's operational ICON analysis and initIFS against ECMWF's operational Integrated Forecasting System (IFS) analysis, respectively, or against direct observations (see Supplementary Table 1 for details on different model configurations). Examples are Figs. 1e, f and 3a. To complement this analysis, Supplementary Fig. 3 shows biases, RMSE and CRMSE for top-of-atmosphere and infrared (10.8 μm) brightness temperatures from the European

Organisation for the Exploitation of Meteorological Satellites (EUMETSAT) Spinning Enhanced Visible and Infrared Imager (SEVIRI) instrument[37]. The original radiances were converted to brightness temperatures following EUMETSAT's guideline and are compared to the model equivalent of 10.8 μm brightness temperature, calculated with the radiative transfer model RTTOV[38] on an hourly basis. Scores in Supplementary Fig. 3 are averaged over August 2016, July and August 2017 because model equivalent brightness temperatures for July 2016 are not available due to technical issues. We suppose that the main conclusions would not change by including July 2016 to the analysis.

Time averaging of the RMSE considers only time steps between which the autocorrelation function of the differences between EXPLC's and PARAM's RMSE has decreased below a critical value. This ensures to only use time steps that are independent from each other, which typically is the case after 18–30 h for data in Fig. 3b–d.

For the evaluation against radiosondes, we use soundings from Abidjan, Ouagadougou, Niamey and Tamanrasset from the Integrated Global Radiosonde Archive (IGRA). These are the only stations in the region of interest with data on at least 40% of the days in the analysed 4-month period (00 and/or 12 UTC). The few other stations in West Africa such as Dakar have less data available and are therefore not considered. The bias with respect to radiosondes is calculated according to Eq. (1), with $x^f$ being the forecast values of 20 different pressure levels at the grid point containing the station, $x^a$ being the sounding values, $n$ the number of available soundings and $w_i = 1$. Results for relative humidity and forecast days 2–5 are shown in Fig. 2. Supplementary Figure 4 shows the corresponding plots for the additional parameters specific humidity, temperature and wind speed. In order to document the temporal evolution of biases against radiosonde observations, Supplementary Figs. 5 and 6 show the results corresponding to Fig. 2 and to Supplementary Fig. 4, respectively, for the forecast range 6–10 days.

**Quantification of the two-way nesting impact**. The DTE, according to Eq. (4), is calculated as an objective metric for the differences between the PARAM and EXPLC simulations:

$$\text{DTE} = \frac{1}{2}\left(\Delta u^2 + \Delta v^2 + \kappa \Delta T^2\right), \qquad (4)$$

where $\kappa = \frac{c_p}{T_r}$, with $c_p = 1005\ \text{J kg}^{-1}\ \text{K}^{-1}$ the specific heat capacity of air and $T_r = 287\ \text{K}$ a reference air temperature[39,40]. $\Delta u$, $\Delta v$ and $\Delta T$ are the differences in zonal and meridional wind speed and in temperature between EXPLC and PARAM. Tests with additionally including latent heat in the DTE computation showed minor differences. The evolution of DTE in latitudinal and longitudinal direction is depicted in Fig. 4 and in Supplementary Fig. 7.

## Data availability

The ICON model output generated for this study is available from the corresponding author upon reasonable request.

Operational ICON analysis data used for model initialisation and evaluation are accessible for registered users via DWD's "PArallel MOdel data REtrieve from Oracle databases" (PAMORE) service at www.dwd.de/DE/leistungen/pamore/pamore.html.

TRMM rainfall, IGRA radiosonde and SEVIRI satellite data sets are available from https://doi.org/10.5067/TRMM/TMPA/3H/7, https://www.ncdc.noaa.gov/data-access/weather-balloon/integrated-global-radiosonde-archive and https://www.eumetsat.int/website/home/Data/DataDelivery/EUMETSATDataCentre/index.html, respectively.

IFS operational analysis data used for additional simulations presented in the Supplementary Information are available via ECMWF's MARS archive at https://apps.ecmwf.int/auth/login/.

## Code availability

The ICON (ICON atmosphere) source code will be made available under the ICON Software License Agreement ISLA version 2.1 (not yet available), which will be a common SLA of the German Weather Service (DWD) and the Max Planck Institute for Meteorology (MPI-M) (www.mpimet.mpg.de/en/science/models/license/).

Data processing scripts are available from the corresponding author upon reasonable request.

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

## Acknowledgements

The research leading to these results has received funding from the European Union 7th Framework Programme (FP7/2007-2013) under Grant Agreement No. 603502 (EU project DACCIWA: Dynamics-aerosol-chemistry-cloud interactions in West Africa) and were conducted in the framework of the Helmholtz Research Programme "Atmosphere and Climate". We gratefully acknowledge Andreas H. Fink for fruitful discussions about meteorology in (sub-)tropical West Africa. We thank Daniel Reinert, Jürgen Helmert and Florian Prill from DWD for their assistance with the ICON model. Simulations were performed on the ForHLR II high-performance computer of the Steinbuch Centre for Computing at KIT. We acknowledge support by the KIT-Publication Fund of the Karlsruhe Institute of Technology.

## Author contributions

G.P. performed all simulations and analysis steps, while P.K. initially formulated the idea for this study. Both authors jointly discussed the results and wrote the paper.

## Additional information

**Competing interests:** The authors declare no competing interests.

