## [Peer Review File · Nature Communications]

Reviewer #1

First review of the manuscript “**Resolving Sahelian thunderstorms improves weather forecasts for Europe**” submitted to *Nature Communications*.

This is a very interesting study showing that increasing resolution over West Africa, using a two-way nesting and turning off convective parameterization significantly increases the 8-10 days forecast skills over Europe. These forecast improvements are investigated by looking at teleconnections: drying above the Sahara appears to be a key driver, impacting the Saharan heat low, and subsequently the African easterly waves and European weather.

The manuscript is well written, and the possible impacts of the results makes it suitable for publication in *Nature communications*. The figures in the main article and in the supplement are of good quality and do support the results very well.

However, I have two major comments (cf. hereafter): to address the first of them, **additional simulations may be needed in order to discriminate between what is gained from increasing resolution and what comes from turning off convective parameterization**. The other one is about **the degradation of forecast skills over the North Atlantic, and how robust can be the forecast improvement in Europe if it is associated with poorer forecast skills in the North Atlantic**. I have also several minor comments, suggestions, and questions listed hereafter for the authors’ consideration.

Major comments

1. The authors compare simulations with 6.5 km non-parameterized convection over West-Africa (using 2-way nesting) with the control simulation using 13 km parameterized simulations globally (including West-Africa, with 2-way nesting only over Europe). At 13 km some of the convection is resolved and the gain of using convection parameterization in MCS regions is unclear and can even be detrimental, for instance for the diurnal cycle of precipitation (White et al. 2018). Hence (unless it has already been done) I think it is essential the authors provide simulations using 2-way nesting over West Africa keeping the “low” horizontal resolution of 13 km but turning off convection parameterization. Beyond the present study, this could also lead to a discussion of possible model improvement including local changes not only in resolution but also in physical parameterization, with the possible risk of regionally over-tuning models, including discontinuities, etc. (problems likely to affect climate simulations).
2. The improvement of the forecast skills over Europe is done at the expense of poorer skills in the Northern Atlantic. How robust can be the forecast improvement in Europe if the North Atlantic does not see an improvement or at least no degradation of forecast quality? (cf. also related minor comments hereafter)

Minor comments

3. L6 (abstract): it is highlighted that organized convection is “rare”, however it is quite

common indeed in West Africa but also in other regions (e.g. US Great Plains, Congo Basin, Maritime continent). I suggest deleting the word “rare” from the abstract (or possibly include a discussion of MCS frequency worldwide in order to explain, in the introduction, in what sense they are “rare”)

4. L 23: recent work of Reinares Martinez and Chaboureau (2018) also supports this point.

5. L 27: replace “with” by “at”

6. l 32-33: This should be rephrased (an improvement over Europe after a few days of simulations? The last sentence is also unclear, I suppose it means that the additional observations were only available during the campaign and hence cannot be use for operational weather forecast.)

7. L36 - 37: I may be wrong but from my understanding ICON allows for refining grid within a single simulation, and hence it is not necessary to perform a 2-way nesting in the classical sense. If you indeed used ICON specific capability to have locally refined grid within one simulation, then this should be explicitly stated (by rephrasing the paragraph l 34-37)

8. L41: Replace “enabling explicit convection” by “turning off the convection parameterization”, as grid-box scale (explicit) convection can also occur at 13 km resolution.

9. L 44-45: depending on how my major comment 1. is addressed question (i) may have to be rephrased

10. Figure 1: what are the 31 simulations? Explanations on how the ensembles are constructed should be included, at least in the supplement.

11. l 55: This is a very broad statement, is it only supported by figure 2 or are there other evidence of circulation improvement over the all troposphere? I think the sentence could be removed or rephrased (or at the very least moved to the end of the paragraph, after the discussion of the figures).

12. L63: is it likely to happened in the future for instance at DWD? Would that be a recommendation of thus study for future operational use of the refined grid over West Africa? IFS initialization (Fig S2) seems to avoid the short-term poorer performances, is it more realistic than the DWD one (although not using 2-way nesting)? Supplement section S3 and figure S2 mention briefly the good performances of IFS initialization, and discuss the impact on the SHL, but without addressing directly the lack of initial shock with IFS initialization.

13. L 68: I agree, although estimating climate prediction skills from a set of 10-day forecasts seems difficult, as feedbacks may cancel the improvements. Following my

major comment 2., the degraded forecast skills over the North Atlantic might also lead to overall poorer (longer-term) forecast quality.

A very interesting extension this study would be to perform seasonal forecasts or even climate simulations with the increased resolution over West Africa (but this may not be CPU affordable yet).

14. L70-75 and Figure 3a: The regions with poorer forecast should also be mentioned and discussed. The focus of the paper is on forecast skills over Europe, however a succinct discussion of the forecast skills globally would be very welcome (possibly showing the whole Earth).

Recalling my major comment 2: Why are the skills degraded over the North Atlantic? Is it likely to also lead to (sometimes, in another forecast) decreased skills over Europe? The forecast shown is for late July 2016, have you also performed forecasts for other years / months to test further the robustness of the results?

15. L78-79: There are arguably many regions in the world more affected by weather changes than Europe both in term of casualties related to extreme events and economic cost. So “the largest” should be replace by “large” (or larger than the other strongly teleconnected region located in the Southern Hemisphere), or the statement could be simply removed. I think there is no need to justify the choice of Europe here (this is the focus of the paper, clearly stated in the title), but as I mentioned before the impact in other regions of the world could also be briefly discussed.

16. L 83: refer the reader to the supplement here (as the operational ICON forecast have not been introduced yet)

17. L 85-87: visible only from day 7 (fig 3b, unless this statement is based on another figure, in which case it should be referred to)

18. L 103-104 (and supplement 1 59-64): Why is latent heat not included in the DTE? If this metric has been used before, the relevant literature should be cited.

19. L 105: Can you justify the choice of this averaging domain $40^{\circ}\text{W} - 0^{\circ}\text{W}$? (it is to focuss on the central and eastern North-Hemisphere Atlantic I suppose, and stills include the Western part of the African domain as well). Plotting the domains discussed here on a map would also be helpful (e.g. on Fig 3a or on Fig S1)

20. L 127-165 and figure 4d and S4: An interesting discussion on how resolving more convective clouds is provided here. However, I think this could be even deepened to investigate (or at least discuss) how a better representation of *organized* convection leads to a drying over the Sahara (as well as to an overall drying, as shown Fig 4d and Fig S4)? Idealized simulations (and hence with much more simple situations) looking at convective organization, its effect on tropospheric humidity and clouds, and the subsequent feedbacks (There is an abundant litterature on that topic, cf. e.g. Mauritsen and Stevens 2015; Mapes 2016; Wing et al. 2018) have shown that there is usually a drying of the surrounding non-convective areas, as well as an overall drying of the whole

domain, when convection becomes more aggregated. It would be very interesting to make a link with these studies (although it might be challenging due to the complex dynamics in play in West Africa compared to idealized simulations), and it might be a way to extend the “discussion” section, which in its current form is more a “conclusion”.

21. 1 136: is this decrease in cloud cover an improvement (compared with observations)?

22. 1 146: It would also be interesting to plot the vertical velocity (and not only the difference). Following my previous comment on convective organization, do you think this drying is linked to a shift in simulations from un-organized convection to MCS? And if yes, how?

23. 1 166: Unless proper discussion is added, replace “discussion” by “conclusion” (or possibly “conclusions and perspectives”)

24. L170: replace “technique” by “approach” (the novelty here is to use a 2-way nesting domain over Africa in addition to the one over Europe, this is not the nesting technique itself)

In the supplement:

25. Supplement 16-8: following my major comment 1., to actually have “only one change” resolution should be kept at 13 km, so that the only change is actually turning off convection parameterization.

26. Fig S1 or fig 3 (in the main text) could include extra boxes e.g. the 2-way nesting domain over Europe, as well as possibly the bands 40-0°W and 8-18°N used fig 4 (consider using dashed boxes to make it still readable)

27. Fig S4 b-e: I suppose the blue lines are the differences between EXPLC and PARAM of the respective setups? The authors should consider making the caption even more explicit (because this is slightly confusing at first as the figures b-e are otherwise showing differences between model versions)

28. Supplement L 106-107: Replace by “more than 15%” as only v2.0.12 initIFS shows reduction exceeding 20% (at 24-25 °N, ~ 450 hPa). Also replace 500 hPa by 450 hPa, the largest reduction being clearly above 500 h Pa (around 425-450 hPa)

29. Supplement 1 110-111: add “over the Sahara” (or “North of 15°N”)

Bibliography

Mapes, B. E., 2016: Gregarious convection and radiative feedbacks in idealized worlds. *J. Adv. Model. Earth Syst.*, **8**, 1029–1033, doi:10.1002/2016MS000651.

- Mauritsen, T., and B. Stevens, 2015: Missing iris effect as a possible cause of muted hydrological change and high climate sensitivity in models. *Nat. Geosci.*, **8**, 346–351, doi:10.1038/ngeo2414.
- Reinares Martínez, I., and J.-P. Chaboureau, 2018: Precipitation and Mesoscale Convective Systems: Explicit versus Parameterized Convection over Northern Africa. *Mon. Weather Rev.*, **146**, 797–812, doi:10.1175/MWR-D-17-0202.1.
- White, B. A., A. M. Buchanan, C. E. Birch, P. Stier, and K. J. Pearson, 2018: Quantifying the Effects of Horizontal Grid Length and Parameterized Convection on the Degree of Convective Organization Using a Metric of the Potential for Convective Interaction. *J. Atmospheric Sci.*, **75**, 425–450, doi:10.1175/JAS-D-16-0307.1.
- Wing, A. A., K. Emanuel, C. E. Holloway, and C. Muller, 2018: Convective Self-Aggregation in Numerical Simulations: A Review. *Shallow Clouds, Water Vapor, Circulation, and Climate Sensitivity*, R. Pincus, D. Winker, S. Bony, and B. Stevens, Eds., *Space Sciences Series of ISSI*, Springer International Publishing, Cham, 1–25 https://doi.org/10.1007/978-3-319-77273-8_1.

Reviewer #2 (Remarks to the Author):

The authors discuss the potential for improved summer weather forecasts over Europe by improving the representation of Sahelian rainfall. To justify their claim they use the global ICON weather forecast model with parametrized convection and compare it to runs with an additional nested domain over Sahel with a convection permitting version of ICON. Daily forecasts for one month are used with different versions of ICON and with initial conditions from ICON assimilation system and ECMWF. The authors then evaluate these runs and discuss typical teleconnections via the Jet streams.

Overall, the subject is well posed, very interesting to the community and clearly shows the potential for significant improvements in European forecasts and therefore should be published. However, while the structure of the manuscript can remain essentially untouched it requires major revision in the sense that the strong claim the authors make can not yet be supported by the results as the simulation period is too short and the uncertainty due to that is too large - Please note that the differences only appear beyond day 8 but there the uncertainty in the deterministic forecasts is increasingly large and this is particularly true for European summer forecasts. Also one should keep in mind that at 6 km resolution convection is only very poorly resolved and the errors are supposed to be large, in particular the activity and energy of convective systems tends to be overestimated.

I therefore ask for the following major revisions:

(1) Please extend the simulation period, it would be sufficient to do this only for your baseline run by adding at least another month for the same year or even better having two years with a June/July period. Ideally, an ensemble run would be even better but it should do without it

(2) In Figure 1d the diurnal cycle from param doesn't look very good. Does the operational version of ICON at that time use

icapdcycl >=2, ie with the diurnal cycle parametrization option? If not it would be fair and useful to also look at year 2017 in order not to overstate the results and overemphasize the differences

(3) The discussion on "changes over Africa" is not clear. Do you suggest that it is mainly due to differences in radiative forcing? If yes make it clear, and do you know that the radiative forcing in the explicit run is better (e.g. from comparisons with Brightness temperatures from Meteosat geostationary) cause it has more clear-sky (convectively inactive) regions?

Also, are you sure that missing cold pools "are responsible for the underestimation of the ventilation of the SHL" in the parametrized run? Or is it just an underestimation of convective activity there and

or teleconnection due to insufficient northward propagation of convective activity in the evening/night-time ? Please double check

(4) Showing geopotential as a metric over the tropics is not useful, eg rmse would essentially be an integrated temperature bias. I suggest to use in Figure 2, either 850 hPa or 200 hPa temperature rmse and/or 850 or 200 hPa wind

Specific comments:

(1) lines 64-66 "the convection parameterisation deteriorates the forecasts during the first days" this sentence is clearly untenable, delete.

(2) lines 98-100 "the shift by 20 hours is comparable to improvements of ECMWF over one decade". Please delete , as you can't compare improvements over one very small domain over a short period - that might be not statistically sufficiently significant" with improvements using very large samples over a hemisphere using a 6-12 months running window

Peter Bechtold

First review of the manuscript “Resolving Sahelian thunderstorms improves weather forecasts for Europe” submitted to Nature Communications.

We thank both Reviewers for their clear and very thoughtful comments, which helped us to substantially improve our manuscript. Apologies for the long time this has taken, but both extensive additional model simulations (three more months) and new comparisons with observations (radiosondes, satellite data) took significant amounts of time.

As a consequence of the revision the paper has changed in many ways but we believe that the core of the message of the original submission still holds, i.e. a statistically significant forecast improvement over Europe in the simulations without convection parameterisation over West Africa (see our new Fig. 3b). In response to the new results, we have somewhat shifted our attention away from Europe to other parts of the globe (as also requested by reviewer 1). Relatively large signals (and some forecast improvements) are also seen in large parts of the tropical belt and in the southern hemispheric extratropics (see our new Fig. 3a). We therefore decided to change the title to now read “Can resolving Sahelian thunderstorms improve global weather forecasts?”.

In addition, and quite importantly we think, we have now identified three key areas where our results suggest ways to further reduce the large errors over Africa that deteriorate long-term forecasts almost globally: (1) Use explicit convection to generate initial conditions over Africa, as those generated with parameterised convection have large biases, (2) repeat our experiments with even higher resolution to avoid problems with more localised convection over southern West Africa, (3) expand the study to more than four months to clearer bring out improvements relative to the large background weather noise in the extratropics.

Here is a short summary of other changes we made:

- *All revised figures in the main article cover four months.*
- *Figure 1 additionally shows horizontal maps of precipitation biases. Overall behaviour is very similar to original version.*
- *Figure 2 was replaced by a direct evaluation with radiosonde observations, as a comparison with analysis data, which is produced using parameterised convection, is somewhat misleading.*
- *Figure 3 shows an almost global plot of impact on forecasts (panel a) in line with our change of strategy outlined above, as well as a simplified analysis of forecast improvement over Europe (panel b).*
- *The Supplementary Information now contains additional analysis of observations and was streamlined for better readability.*

Especially changes to figures 2 and 3 make the paper much stronger and clearer we think.

Please find our replies in red italics below.

REVIEWER 1

This is a very interesting study showing that increasing resolution over West Africa, using a two-way nesting and turning off convective parameterization significantly increases the 8-10 days forecast skills over Europe. These forecast improvements are investigated by looking at teleconnections: drying above the Sahara appears to be a key driver, impacting the Saharan heat low, and subsequently the African easterly waves and European weather.

The manuscript is well written, and the possible impacts of the results makes it suitable for publication in Nature communications. The figures in the main article and in the supplement are of good quality and do support the results very well. However, I have two major comments (cf. hereafter): to address the first of them, additional simulations may be needed in order to discriminate between what is gained from increasing resolution and what comes from turning off convective parameterization. The other one is about the degradation of forecast skills over the North Atlantic, and how robust can be the forecast improvement in Europe if it is associated with poorer forecast skills in the North Atlantic. I have also several minor comments, suggestions, and questions listed hereafter for the authors’ consideration.

Major comments

1. The authors compare simulations with 6.5 km non-parameterized convection over West-Africa (using 2-way nesting) with the control simulation using 13 km parameterized simulations globally (including West-Africa,

with 2-way nesting only over Europe). At 13 km some of the convection is resolved and the gain of using convection parameterization in MCS regions is unclear and can even be detrimental, for instance for the diurnal cycle of precipitation (White et al. 2018). Hence (unless it has already been done) I think it is essential the authors provide simulations using 2-way nesting over West Africa keeping the “low” horizontal resolution of 13 km but turning off convection parameterization. Beyond the present study, this could also lead to a discussion of possible model improvement including local changes not only in resolution but also in physical parameterization, with the possible risk of regionally over-tuning models, including discontinuities, etc. (problems likely to affect climate simulations).

We performed an additional simulation with the parameterisation switched off at 13 km grid spacing over Africa. Since grid spacings for nests in ICON have to be at least a factor of two finer than in the parent domain, we had to nest a 13 km domain over West Africa into a 26 km global domain rather than using 13 km everywhere. The figure below shows Hovmoeller diagrams of precipitation similar to Fig. 1a–c in the main paper. It clearly demonstrates that, indeed, 13 km grid spacing without parameterisation (middle panel) is able to reproduce single convective systems much more realistically (as compared to TRMM measurements, right panel) than the 26 km (left panel) and the 13 km (Fig. 1a in original manuscript) global simulations with parameterisation. We added the figure below to the Supplementary Information and also a respective comment to the main text.

In addition, our more detailed analysis of rainfall fields revealed that over the Niger Delta and the Guinea Highlands the model faces problems with convective activity even with the 6.5 km grid spacing. In these regions, convection is less organized than in the Sahel and without the parameterisation rainfall is underestimated. To document this issue, we added two panels to Fig. 1 and a short discussion to the text. Despite this, we see large-scale improvement in the moisture bias over West Africa with explicit convection (see new Fig. 2). It would be desirable to run even higher resolution to test possible improvements over this southern region, but this is beyond the CPU time we have available for this study. We added a comment on this to the conclusion section.

2. The improvement of the forecast skills over Europe is done at the expense of poorer skills in the Northern Atlantic. How robust can be the forecast improvement in Europe if the North Atlantic does not see an improvement or at least no degradation of forecast quality? (cf. also related minor comments hereafter)

As suggested by Reviewer 2, we performed additional simulations with our baseline setup for three more months (August 2016, July and August 2017). Averaging over now four months, we find even more wave-like structures in the RMSE fields (see revised Fig. 3a). It confirms that the changed atmospheric situation over West Africa induces signals that propagate along wave guides into the extratropics of both hemispheres. While signals in the wider tropics are rather smooth, the more variable extratropical stormtrack regions show a certain level of noise.

iness. Only through regarding regional averages (e.g. that over Europe) can we bring out a statistically significant forecast improvement. We expect that a study covering many more months would be able to average out some of the noise we see here, but due to CPU restrictions such an analysis is beyond the scope of this study. We added respective comments and a more balanced description to the text.

Minor comments

3. L6 (abstract): it is highlighted that organized convection is “rare”, however it is quite common indeed in West Africa but also in other regions (e.g. US Great Plains, Congo Basin, Maritime continent). I suggest deleting the word “rare” from the abstract (or possibly include a discussion of MCS frequency worldwide in order to explain, in the introduction, in what sense they are “rare”)

We decided to replace “rare” with “special”.

4. L 23: recent work of Reinares Martinez and Chaboureau (2018) also supports this point.

Good suggestion, added.

5. L 27: replace “with” by “at”

Done.

6. l 32-33: This should be rephrased (an improvement over Europe after a few days of simulations? The last sentence is also unclear, I suppose it means that the additional observations were only available during the campaign and hence cannot be use for operational weather forecast.)

First sentence now reads “Indeed, a forecast improvement over Europe after some days leadtime could be revealed”. Last sentence was omitted as suggested and “wet monsoon season 2006” added to previous sentence to indicated the limited availability of data.

7. L36 - 37: I may be wrong but from my understanding ICON allows for refining grid within a single simulation, and hence it is not necessary to perform a 2-way nesting in the classical sense. If you indeed used ICON specific capability to have locally refined grid within one simulation, then this should be explicitly stated (by rephrasing the paragraph l 34-37)

This is a misunderstanding. ICON’s dynamical core is very flexible to introduce subdomains with higher resolution and to exchange information between the two. Whether you prefer to call this “single simulation with re-fined grid” or “two-way nesting” is not entirely obvious. In the ICON community the latter is more common and therefore we prefer to use that. We slightly rephrased the respective paragraph to make this clearer.

8. L41: Replace “enabling explicit convection” by “turning off the convection parameterization”, as grid-box scale (explicit) convection can also occur at 13 km resolution.

Changed as suggested.

9. L 44-45: depending on how my major comment 1. is addressed question (i) may have to be rephrased

As the negative impact of the convection parameterisation appears to be the main problem here, we would like to leave this point as is.

10. Figure 1: what are the 31 simulations? Explanations on how the ensembles are constructed should be included, at least in the supplement.

We added the following information to the main text: “For both PARAM and EXPLC we performed in total 124 10-day forecasts initialised every day in July and August 2016 and 2017 at 12 UTC.”

11. l 55: This is a very broad statement, is it only supported by figure 2 or are there other evidence of circulation improvement over the all troposphere? I think the sentence could be removed or rephrased (or at the very least moved to the end of the paragraph, after the discussion of the figures).

Fair point. As we have now replaced Fig. 2 with a comparison with radiosondes that do show a more ambiguous impact of explicit convection, this whole section is changed and now gives a fairer account of the differences.

12. L63: is it likely to happen in the future for instance at DWD? Would that be a recommendation of this study for future operational use of the refined grid over West Africa? IFS initialization (Fig S2) seems to avoid the short-term poorer performances, is it more realistic than the DWD one (although not using 2-way nesting)? Supplement section S3 and figure S2 mention briefly the good performances of IFS initialization, and discuss the impact on the SHL, but without addressing directly the lack of initial shock with IFS initialization.

As far as we know, there are no concrete plans in this direction but yes, it is in fact a recommendation from this work that we have now explicitly integrated into the conclusion section. A new aspect that further supports this idea is the large moisture bias that appears to be gradually easing with explicit convection.

Due to the enhanced emphasis on the comparison with observations and the changes to Figs. 2 and 3, we are not discussing the problem of an initial shock over West Africa anymore, as we think the moisture bias evident from the radiosondes may be even more significant.

With respect to initialisation, we should first point out that all runs shown in the old Fig. S2 and S4 compare two-way nesting with global only simulations in a consistent way. What changes is the model version, the initialisation data and whether or not a nesting over Europe is used. It is true that the runs initialised with IFS all show a relatively good performance but clearly the model version is also quite important for the response. At the end of the day, the main point here is to show that the results are generally sensitive to details of the configuration rather than going into more detail about the exact reasons, particularly as those may not be statistically significant for only one month of simulation available for this analysis.

In response to this point, we decided to drop the old Fig. S2, also as it is not analogous to Fig. 3b in the main paper anymore, and to develop the discussion around Table S2 only. The respective text passage was streamlined accordingly.

13. L68: I agree, although estimating climate prediction skills from a set of 10-day forecasts seems difficult, as feedbacks may cancel the improvements. Following my major comment 2., the degraded forecast skills over the North Atlantic might also lead to overall poorer (longer-term) forecast quality. A very interesting extension this study would be to perform seasonal forecasts or even climate simulations with the increased resolution over West Africa (but this may not be CPU affordable yet).

100% agreed. We have removed this aspect from this part of the paper and included it a little more clearly into the outlook at the very end (conclusions and perspectives).

14. L70-75 and Figure 3a: The regions with poorer forecast should also be mentioned and discussed. The focus of the paper is on forecast skills over Europe, however a succinct discussion of the forecast skills globally would be very welcome (possibly showing the whole Earth). Recalling my major comment 2: Why are the skills degraded over the North Atlantic? Is it likely to also lead to (sometimes, in another forecast) decreased skills over Europe? The forecast shown is for late July 2016, have you also performed forecasts for other years / months to test further the robustness of the results?

Given the new results for the four (instead of only one) months, we have adapted our presentation and the discussion of the results. We changed the title to now read "Can resolving Sahelian thunderstorms improve global weather forecasts?". We then discuss three regions in the text based on Fig. 3a: the tropics, the southern midlatitudes and the northern midlatitudes, with the latter two both showing the noisiness you describe above. Our conclusion here is that the noisiness of the midlatitude stormtrack is too strong to show a clear improvement everywhere after only 4 months of investigation time. However, if you average over larger regions such as a European box, you get a positive impact for all parameters and levels, but the magnitude is only 1-2% and not all parameters are statistically significant (see new Fig. 3b). We also now express this in the conclusions with a recommendation to operational services to explore this further using re-forecasts for more months.

15. L78-79: There are arguably many regions in the world more affected by weather changes than Europe both in term of casualties related to extreme events and economic cost. So "the largest" should be replaced by "large" (or larger than the other strongly teleconnected region located in the Southern Hemisphere), or the statement could be simply removed. I think there is no need to justify the choice of Europe here (this is the focus of the paper, clearly stated in the title), but as I mentioned before the impact in other regions of the world could also be briefly discussed.

Good point. We replaced "larger" by "large" and also now discuss other regions, as pointed out above.

16. L 83: refer the reader to the supplement here (as the operational ICON forecast have not been introduced yet) *This figure (and the associated text) was redesigned and does not contain the operational ICON forecast anymore.*

17. L 85-87: visible only from day 7 (fig 3b, unless this statement is based on another figure, in which case it should be referred to)

This figure (and the associated text) was redesigned and does not contain information on individual forecast days anymore.

18. L 103-104 (and supplement l 59-64): Why is latent heat not included in the DTE? If this metric has been used before, the relevant literature should be cited.

This goes back to a paper on mesoscale predictability by Zhang et al. (2003), which was already cited in the Supplement. We think that this should suffice and therefore did not make any changes.

At some point, we did some experiments with including qL in DTE and found that this is much smaller than the other terms anyway. We therefore added the following sentence to the Supplement “Tests with additionally including latent heat in the DTE computation showed minor differences.”

19. L 105: Can you justify the choice of this averaging domain $40^{\circ}W - 0^{\circ}W$? (it is to focuss on the central and eastern North-Hemisphere Atlantic I suppose, and stills include the Western part of the African domain as well). Plotting the domains discussed here on a map would also be helpful (e.g. on Fig 3a or on Fig S1)

Yes, the idea is to document how the signal leaves Africa with the easterly jet to finally leave an imprint over the eastern Atlantic / western Europe. Some justification for this can be found in the new Fig. 3a, where the Rossby wave train into Europe is evident in the black contours. We added a short comment on this when we introduce Fig. 4a. In addition, we added boxes showing the averaging areas for Fig. 4 in Fig. 3a.

20. L 127-165 and figure 4d and S4: An interesting discussion on how resolving more convective clouds is provided here. However, I think this could be even deepened to investigate (or at least discuss) how a better representation of organized convection leads to a drying over the Sahara (as well as to an overall drying, as shown Fig 4d and Fig S4)? Idealized simulations (and hence with much more simple situations) looking at convective organization, its effect on tropospheric humidity and clouds, and the subsequent feedbacks (There is an abundant litterature on that topic, cf. e.g. Mauritsen and Stevens 2015; Mapes 2016; Wing et al. 2018) have shown that there is usually a drying of the surrounding non-convective areas, as well as an overall drying of the whole domain, when convection becomes more aggregated. It would be very interesting to make a link with these studies (although it might be challenging due to the complex dynamics in play in West Africa compared to idealized simulations), and it might be a way to extend the “discussion” section, which in its current form is more a “conclusion”.

Excellent point, thank you. Now that we have a comparison with radiosonde humidity observations, we decided to bring this point up in the discussion of Fig. 2. In fact, ICON PARAM (and thus the analysis used for initialisation) has an enormous moist bias that is slowly reduced in EXPL. We also included the seminal papers you suggested.

21. l 136: is this decrease in cloud cover an improvement (compared with observations)?

As discussed above, we have now increased the comparison with observations in Figs. 1 and 2. This has revealed that while conditions over the Sahel are clearly improved (rainfall, radiosondes, satellite brightness temperatures), the situation is not as clear cut over southern West Africa, where we suspect the 6.5-km grid spacing to be insufficient to trigger enough convection. Therefore the decrease in cloud cover appears to be an improvement over the Sahel (at least according to the top-of-atmosphere infrared brightness temperatures, see Supplementary Information), while farther south it is not. We added the following sentence to the discussion: “Recall from the earlier discussion that this implies a better agreement with satellite data over the Sahel, but a worse agreement farther south (Fig. S3).”

22. l 146: It would also be interesting to plot the vertical velocity (and not only the difference). Following my previous comment on convective organization, do you think this drying is linked to a shift in simulations from un-organized convection to MCS? And if yes, how?

Yes, we believe that the tendency for more organisation in EXPLC creates a higher precipitation efficiency. Clouds have less interfaces with the ambient air to exchange water vapour and condensate through entrainment/detrainment. This point is mentioned in the paragraph that follows immediately and therefore we feel that there is no need for a change. As Fig. 4 is already rather full, we do not want to include absolute vertical velocity here. We are also not sure that we would learn much more from the absolute values, as these will surely show the general structure of the monsoon circulation known from literature.

23. l 166: Unless proper discussion is added, replace “discussion” by “conclusion” (or possibly “conclusions and perspectives”)

Good suggestion, thank you!

24. L170: replace “technique” by “approach” (the novelty here is to use a 2-way nesting domain over Africa in addition to the one over Europe, this is not the nesting technique itself)

Done.

In the supplement:

25. Supplement l6-8: following my major comment 1., to actually have “only one change” resolution should be kept at 13 km, so that the only change is actually turning off convection parameterization.

As pointed out above, we tested this and found quite consistent results over the Sahel. However, in southern West Africa already 6.5km appears to cause some issues and therefore we prefer to use that for the main paper. Nevertheless we have now included the figure shown above in the Supplement and comment on this aspect in the text.

26. Fig S1 or fig 3 (in the main text) could include extra boxes e.g. the 2-way nesting domain over Europe, as well as possibly the bands 40-0°W and 8-18°N used fig 4 (consider using dashed boxes to make it still readable)

Good idea, boxes are now included in Fig. 3a as already mention above.

27. Fig S4 b-e: I suppose the blue lines are the differences between EXPLC and PARAM of the respective setups? The authors should consider making the caption even more explicit (because this is slightly confusing at first as the figures b-e are otherwise showing differences between model versions)

OK, good point, we have re-structured the caption to make this clearer.

28. Supplement L 106-107: Replace by “more than 15%” as only v2.0.12 initIFS shows reduction exceeding 20% (at 24-25 °N, ~ 450 hPa). Also replace 500 hPa by 450 hPa, the largest reduction being clearly above 500 hPa (around 425-450 hPa)

Changed as suggested.

29. Supplement l 110-111: add “over the Sahara” (or “North of 15°N”)

Done.

Bibliography

Mapes, B. E., 2016: Gregarious convection and radiative feedbacks in idealized worlds. *J. Adv. Model. Earth Syst.*, 8, 1029–1033, doi:10.1002/2016MS000651.

Mauritsen, T., and B. Stevens, 2015: Missing iris effect as a possible cause of muted hydrological change and high climate sensitivity in models. *Nat. Geosci.*, 8, 346–351, doi:10.1038/ngeo2414.

Reinares Martínez, I., and J.-P. Chaboureaud, 2018: Precipitation and Mesoscale Convective Systems: Explicit versus Parameterized Convection over Northern Africa. *Mon. Weather Rev.*, 146, 797–812, doi:10.1175/MWR-D-17-0202.1.

- White, B. A., A. M. Buchanan, C. E. Birch, P. Stier, and K. J. Pearson, 2018: Quantifying the Effects of Horizontal Grid Length and Parameterized Convection on the Degree of Convective Organization Using a Metric of the Potential for Convective Interaction. *J. Atmospheric Sci.*, 75, 425–450, doi:10.1175/JAS-D-16-0307.1.
- Wing, A. A., K. Emanuel, C. E. Holloway, and C. Muller, 2018: Convective Self-Aggregation in Numerical Simulations: A Review. *Shallow Clouds, Water Vapor, Circulation, and Climate Sensitivity*, R. Pincus, D. Winker, S. Bony, and B. Stevens, Eds., Space Sciences Series of ISSI, Springer International Publishing, Cham, 1–25 https://doi.org/10.1007/978-3-319-77273-8_1.

REVIEWER 2 (Peter Bechtold)

The authors discuss the potential for improved summer weather forecasts over Europe by improving the representation of Sahelian rainfall. To justify their claim they use the global ICON weather forecast model with parameterized convection and compare it to runs with an additional nested domain over Sahel with a convection permitting version of ICON. Daily forecasts for one month are used with different versions of ICON and with initial conditions from ICON assimilation system and ECMWF. The authors then evaluate these runs and discuss typical teleconnections via the Jet streams.

Overall, the subject is well posed, very interesting to the community and clearly shows the potential for significant improvements in European forecasts and therefore should be published. However, while the structure of the manuscript can remain essentially untouched it requires major revision in the sense that the strong claim the authors make can not yet be supported by the results as the simulation period is too short and the uncertainty due to that is too large - Please note that the differences only appear beyond day 8 but there the uncertainty in the deterministic forecasts is increasingly large and this is particularly true for European summer forecasts. Also one should keep in mind that at 6 km resolution convection is only very poorly resolved and the errors are supposed to be large, in particular the activity and energy of convective systems tends to be overestimated.

I therefore ask for the following major revisions:

(1) Please extend the simulation period, it would be sufficient to do this only for your baseline run by adding at least another month for the same year or even better having two years with a June/July period. Ideally, an ensemble run would be even better but it should do without it.

We performed additional simulations with our baseline setup for three more months (August 2016, July and August 2017). Calculating four-month averages of the RMSE shows a less pronounced signal in terms of the shift of the forecast horizon. This indicates that July 2016 was indeed a month with an extraordinary positive impact on the mid-latitudes (that we selected by chance). On the other hand, data from now 124 initialization days yield statistically more robust results. We therefore show the mean improvement over Europe in terms of RMSE averaged over forecast days 5–8 in the revised Figure 3b. This resonates with your point above that beyond day 8 the uncertainty becomes large and the signal weakens.

(2) In Figure 1d the diurnal cycle from param doesn't look very good. Does the operational version of ICON at that time use `icapdcycl >=2`, ie with the diurnal cycle parametrization option? If not it would be fair and useful to also look at year 2017 in order not to overstate the results and overemphasize the differences.

We use `icapdcycl=3` for all the simulations with convection parameterization switched on, as done operationally at DWD. Some preliminary tests without this switch showed an even earlier peak in parameterized convection around 12 UTC.

(3) The discussion on "changes over Africa" is not clear. Do you suggest that it is mainly due to differences in radiative forcing? If yes make it clear, and do you know that the radiative forcing in the explicit run is better (e.g. from comparisons with Brightness temperatures from Meteosat geo-stationary) cause it has more clear-sky (convectively inactive) regions?

Also, are you sure that missing cold pools "are responsible for the underestimation of the ventilation of the SHL" in the parametrized run? Or is it just an underestimation of convective activity there and or teleconnection due to insufficient northward propagation of convective activity in the evening/night-time ? Please double check.

All very good points. Over the Sahel and Sahara the radiative forcing appears to be better, as now discussed earlier in the paper. A reference to this discussion is now included. Nevertheless we toned down the importance of radiation a little in the discussion, as we have no solid proof of its importance.

The idea with the cold pools comes from the literature and not from an explicit analysis provided here. We have now made that clear in the text. The comparison with the radiosonde in Tamanrasset is also consistent with more cold pools in EXPLC but unfortunately that same comparison also reveals other problems, likely to do with boundary layer parameterisation. A reference to this comparison has now been included.

(4) Showing geopotential as a metric over the tropics is not useful, eg rmse would essentially be an integrated temperature bias. I suggest to use in Figure 2, either 850 hPa or 200 hPa temperature rmse and/or 850 or 200 hPa wind.

In the original manuscript we indicate that the analysis fields, to which we compare our forecasts, suffer from using the convection parameterization and from the sparseness of observations in West Africa (line 63ff). To prevent this comparison of apples and oranges (parameterized analysis vs explicit forecasts) we show a comparison with radiosondes and satellite data in the revised Figures 1 and 2 (plus Supplement) instead.

Specific comments:

(1) lines 64-66 "the convection parameterisation deteriorates the forecasts during the first days" this sentence is clearly untenable, delete.

Figure was replaced and with it this sentence.

(2) lines 98-100 "the shift by 20 hours is comparable to improvements of ECMWF over one decade". Please delete , as you can't compare improvements over one very small domain over a short period - that might be not statistically sufficiently significant" with improvements using very large samples over a hemisphere using a 6-12 months running window

Fair point. As we now regard reductions in RMSE and not forecast horizon, we had to rephrase. We still include a similar comparison but tried to word it in a way that time-space differences in this comparison are clear.

Reviewer #1 (Remarks to the Author):

Second review of “Can resolving Sahelian thunderstorms improve global weather forecasts?” submitted to Nature Communications.

The authors addressed all of my comments, and carried out extensive work, performing new simulations and changing their evaluation strategy to gain in consistency, leading to a significantly improved manuscript. I still have some issues with the robustness of forecasts improvement over Europe, but I believe this can be addressed mostly by being even more cautious in the interpretations and conclusions. I hence recommend this paper for publication in Nature Communications, providing the authors address my comments hereafter.

General comments:

1. The revised paper is based on 10-day forecasts over 4 months. As the authors mentioned in their response, this is still not long enough to discuss sub-regional features in noisy areas. It is also a limit to the robustness of European forecast improvement, which seems to be driven by a strongly improved forecast over Scandinavia. This should be discussed, and possibly lead to more cautious conclusions of forecast improvements (cf. my specific comment 13 hereafter).
2. The revised title suggests a focus on global weather, while in the paper only one metric (500 hPa geopotential) is discussed globally. To justify the title, I suggest the authors present error scores (like in figure 3b) for other regions than Europe (e.g. the Americas, South Asia, the tropics). Otherwise, I suggest switching back to the initial title.
3. The drying in convection permitting simulations is too strong at least in some areas (cf. my comments 9 and 10 hereafter), especially over longer forecast ranges suggesting it may be related to model errors rather than to initial conditions. This could be commented in the paper, as this is often a shortcoming in global models development: improving (or here removing) a parameterization can have adverse effects, for instance due to some other well-tuned parameters needing re-tuning.
4. The discussion on forecast performances over West Africa shows somehow mixed results, although mostly improved by the use in the explicit convection simulation. I understand the authors' willingness to avoid comparing with analysis because of their use of parameterizations, however not providing them in the discussion weakens the conclusions on forecast performances over West Africa (as objective metrics like the RMSE improvements shown Fig 3b are not provided for West-Africa). Analysis derived from convection-permitting simulations would be very useful for model evaluation too (along with model initialization), and this could be mentioned in the paper.

Specific comments:

5. l56-57: I suggest you show the same time period in Figure S2 and in Figure 1. You could also show in the supplement the average diurnal cycle for 13 km explicit simulations.
6. l60-62: could be better supported (e.g. citing references of higher resolution simulations in this area)
7. l66: replace “literally” by “almost”
8. l76: replace “party” by “partly”, delete “considerable”.
9. l71: add “and the 500 - 250 hPa layer over Ouagadougou”; to be mention in the discussion too.
10. l78 - 79 and Figure S5: Mention / discuss that this now leads to a dry bias in EXPLC in Ouagadougou and Abidjan (~ 600 hPa - 250 h Pa)

11. l79-81: This is an interesting point indeed, but it gets a bit lost in the middle of the discussion of supplementary figures, showing mixed results. I suggest to move this statement to the end of the paragraph /sub-section.

12. l104: delete "exemplarily"

13. l104-105 : From fig 3a, it looks like if you had chosen a smaller box for instance ending at 60N (and hence including only the southern part of Scandinavia, but still including almost all western European population) the results would have been different. This raises again the question of the robustness of your results that were already raised by myself and the other reviewer in the first round. I suggest mentioning this point (possibly adding a figure like Fig 3b for the smaller Europe region) and possibly de-emphasizing the scale of forecasts improvement.

14. l198: replace "explicit forecasts" by "explicit convection forecasts" or "convection permitting forecasts" .

15. l199: better initial conditions and products for further forecast evaluations.

16. In the supplement, l 7-8: Turning off only shallow convection may not be possible with the Tiedtke scheme, however, in cases of fully separated deep and shallow convection schemes such an approach would be interesting at 6.5 km (as it is too coarse to represent most of shallow convection). Is it something you have considered testing in ICON?

Laurent Labbouz.

Reviewer #2 (Remarks to the Author):

I am happy that the authors extended their forecast period and added validation against radiosonde data as suggested. This made the manuscript acceptable..

I just have a few minor revisions concerning the discussion of the moisture structure (bias) that should be addressed

-l71-71: "problems in the model to transport low-level out of the deep Sahara boundary-layer.. overshooting plumes". No, if it was a vertical transport problem you would get a vertical + and - structure or vice versa. Please remove this sentence. You should say instead that this is a large-scale circulation issue and/or response to the heating - by the way you discuss this yourself in l163-164.

-l76: typo, "party"->"partly"

-l84-85: "forecasts improve ..", please specific that it is the "rms" that improves. " .. indicating a better net longwave radiative balance ..". This is not true, at least you do not show net-longwave (maybe the longwave down improves) but as the brightness temperature bias are much larger in EXPL longwave radiative balance cannot be better. Please correct this sentence

-l138 "west- adn eastward ", citing only kelvin waves here is misleading as this implies eastward propagation, while African easterly waves are Rossby-type waves

-l175: remove "entrainment" as entrainment does not change environment directly, only through compensating subsidence

-l191: "improvement of moist bias through enhanced precip efficiency". This maybe, but is not

certain at all, as the large-scale circulation (moisture advection changes) as you said yourself - btw this might also contribute why EXPL gets worse in the southern part

-I204: "should"->"might" (note that global NWP/data assimilation does generally not apply a nesting strategy)

Second review of the manuscript “Can resolving Sahelian thunderstorms improve global weather forecasts?” submitted to *Nature Communications*.

We thank both Reviewers for their again very constructive comments on the revised version of our manuscript. Please find our replies in red italics below and all modifications to the manuscript with track changes enabled.

Beyond the points described below, we changed Fig. 1d. The old version showed the average over all lead times and all simulations for every time of the day with the standard deviation calculated from the 10 different lead times after averaging over all simulations. We felt that this may go against the readers’ intuition of what the shading indicates. For the revised version we therefore calculated the standard deviation from the 124 different simulations after computing the mean diurnal cycle for each of the 10-day forecasts. This ensures to represent typical variations in the diurnal cycle over the 4 months period rather than due to different lead times. This change is rather cosmetic and the conclusions are not affected by this change.

Furthermore, we shortened the abstract and moved sections 1 and 2 from the Supplementary Information into the new Methods section in order to fulfil the formatting guidelines of Nature Communications. Track changes here refer to the previous version of these sections in the supplementary document.

REVIEWER 1 (Laurent Labbouz)

The authors addressed all of my comments, and carried out extensive work, performing new simulations and changing their evaluation strategy to gain in consistency, leading to a significantly improved manuscript. I still have some issues with the robustness of forecasts improvement over Europe, but I believe this can be addressed mostly by being even more cautious in the interpretations and conclusions. I hence recommend this paper for publication in *Nature Communications*, providing the authors address my comments hereafter.

General comments:

1. The revised paper is based on 10-day forecasts over 4 months. As the authors mentioned in their response, this is still not long enough to discuss sub-regional features in noisy areas. It is also a limit to the robustness of European forecast improvement, which seems to be driven by a strongly improved forecast over Scandinavia. This should be discussed, and possibly lead to more cautious conclusions of forecast improvements (cf. my specific comment 13 hereafter).
Following this and your second comment, we extended the evaluation to comprise more and larger domains in Figure 3b–d of the revised manuscript. The newly defined European evaluation domain is much larger (35–80°N, 20°W–60°E) than the old one and therefore yields more robust results (cf. reply to comment 13 below). The improvement is still significant and on the same order of magnitude as in the smaller domain of the previous version of the paper. In our opinion, the larger domain, in combination with the discussion of the noisy patterns, justifies the conclusions we draw about the potential of our modeling strategy to improve medium-range forecasts.
2. The revised title suggests a focus on global weather, while in the paper only one metric (500 hPa geopotential) is discussed globally. To justify the title, I suggest the authors present error scores (like in figure 3b) for other regions than Europe (e.g. the Americas, South Asia, the tropics). Otherwise, I suggest switching back to the initial title.
We agree. In addition to the enlarged European domain, the revised Figure 3 shows the RMSE improvement for a region in the southern hemispheric extratropics and the tropical belt. The changes in all the three regions are now discussed in the section “... and beyond”.
3. The drying in convection permitting simulations is too strong at least in some areas (cf. my comments 9 and 10 hereafter), especially over longer forecast ranges suggesting it may be related to model errors rather than to initial conditions. This could be commented in the paper, as this is often a shortcoming in global models development: improving (or here removing) a parameterization can have adverse effects, for instance due to some other well-tuned parameters needing re-tuning.

Well spotted! We now emphasise that re-tuning is likely to further improve the convection permitting simulations in the last paragraph of the section “Impact of Sahelian MCSs on forecasts over Africa” and in the perspectives section.

4. The discussion on forecast performances over West Africa shows somehow mixed results, although mostly improved by the use in the explicit convection simulation. I understand the authors’ willingness to avoid comparing with analysis because of their use of parameterizations, however not providing them in the discussion weakens the conclusions on forecast performances over West Africa (as objective metrics like the RMSE improvements shown Fig 3b are not provided for West-Africa). Analysis derived from convection-permitting simulations would be very useful for model evaluation too (along with model initialization), and this could be mentioned in the paper.

We fully agree. By adapting the discussion of Fig. 2 as suggested in the specific comment 11 and adding the statement about re-tuning (cf. comment 3), we communicate this point much clearer in the revised version.

Specific comments:

5. 156-57: I suggest you show the same time period in Figure S2 and in Figure 1. You could also show in the supplement the average diurnal cycle for 13 km explicit simulations.

We conducted the simulations with 13 km explicit convection for one initialization date only (5 August 2017). Following your suggestion, we now show this initialization time also in Fig. 1a–c in the revised version and slightly adapted the description in the respective section.

In the supplement we decided not to show the average diurnal cycle to avoid mis-interpretation, because this average would be based on a single 10-day simulation only and therefore it would not show the same as Fig. 1d.

6. 160-62: could be better supported (e.g. citing references of higher resolution simulations in this area)

A very recent paper (April 2019) by Kendon et al. in Nature Communications compares 4.5 km simulations with TRMM and CMORPH observations. They find similar patterns in the rainfall bias w.r.t. to TRMM but also differences between the two observational data sets. We added one sentence citing this paper in the revised version.

7. 166: replace “literally” by “almost”

Done.

8. 176: replace “party” by “partly”, delete “considerable”.

Corrected.

9. 171: add “and the 500 - 250 hPa layer over Ouagadougou”; to be mention in the discussion too.

In the revised version of Fig. 2 (and S4, S5 and S6) we added open markers for all mean bias values if the differences between EXPLC and PARAM are not significant (cf. first comment of Peter Bechtold). Figure 2b now shows that there is no moist bias in PARAM between 250–600 hPa and we changed the sentence accordingly.

10. 178 - 79 and Figure S5: Mention / discuss that this now leads to a dry bias in EXPLC in Ouagadougou and Abidjan (~ 600 hPa - 250 h Pa)

We mention the dry bias in the revised version.

11. 179-81: This is an interesting point indeed, but it gets a bit lost in the middle of the discussion of supplementary figures, showing mixed results. I suggest to move this statement to the end of the paragraph /sub-section.

Changed as suggested.

12. 1104: delete “exemplarily”

Done.

13. 1104-105: From fig 3a, it looks like if you had chosen a smaller box for instance ending at 60N (and hence including only the southern part of Scandinavia, but still including almost all western European population) the results would have been different. This raises again the question of the robustness of your results that were already raised by myself and the other reviewer in the first round. I suggest mentioning this point (possibly adding a figure like Fig 3b for the smaller Europe region) and possibly de-emphasizing the scale of forecasts improvement.

Showing a smaller box can easily be misleading and hard to interpret because small changes to the borders might change the results substantially – as you write. We therefore decided to do the opposite and to increase the evaluation domain (revised Fig. 3b) in order to yield more robust results. The spatially averaged error scores of the enlarged domain are less sensitive to differences in only small parts of the domain, because they get less weight.

Some additional background information on how we chose the domain in the first place:

When we started this work, we focused our evaluation on the European/North African domain as defined by the WMO for “standardized verification of NWP products”. We then moved the southern border from 25°N to 35°N in order to prevent the boundary zone of the nesting domain to be part of the evaluation.

Such regions are also used by operational weather services (e.g. German weather service) when deciding if changes in the model should become part of the operational setup. Changes in the model setup or new developments will most likely not lead to smooth and only beneficial changes in such horizontal map plots of error scores due to the chaotic nature of the atmosphere as described in the manuscript. By enlarging the European domain (revised Fig. 3b) we increase the robustness of the spatially averaged error scores because very strong differences in only small parts of the domain get a smaller weight. As mentioned in our reply to the first comment, this justifies the conclusions we draw from this analysis.

14. 1198: replace “explicit forecasts” by “explicit convection forecasts” or “convection permitting forecasts”.
Changed to “convection permitting forecasts”.

15. 1199: better initial conditions and products for further forecast evaluations.
We added “and products for model evaluation” here.

16. In the supplement, 17-8: Turning off only shallow convection may not be possible with the Tiedtke scheme, however, in cases of fully separated deep and shallow convection schemes such an approach would be interesting at 6.5 km (as it is too coarse to represent most of shallow convection). Is it something you have considered testing in ICON?

It is possible to turn off only the shallow convection parameterisation in current ICON versions. However, in version 2.0.01, which we used for our first simulations for this work, this was not possible and we stayed with this configuration even when changing to a newer model version. We now mention this as one aspect in future tests in the perspectives.

REVIEWER 2 (Peter Bechtold)

I am happy that the authors extended their forecast period and added validation against radiosonde data as suggested. This made the manuscript acceptable.

I just have a few minor revisions concerning the discussion of the moisture structure (bias) that should be addressed

- 171-71: "problems in the model to transport low-level out of the deep Sahara boundary-layer.. overshooting plumes". No, if it was a vertical transport problem you would get a vertical + and - structure or vice versa. Please remove this sentence. You should say instead that this is a large-scale circulation issue and/or response to the heating - by the way you discuss this yourself in 1163-164.

There actually is such a dipole (too moist boundary layer – too dry above). From the old version of Fig. 2d this was maybe not clear enough because the differences between PARAM and EXPLC are not significant and therefore no markers were plotted. In the revised figure we marked all mean values with open markers if the differences are not significant. This now clearly shows this dipole. Open markers are also added for Figures S4, S5 and S6.

-176: typo, "party" -> "partly"

Corrected.

-184-85: "forecasts improve ..", please specific that it is the "rms" that improves. " .. indicating a better net longwave radiative balance ..". This is not true, at least you do not show net-longwave (maybe the longwave down improves) but as the brightness temperature bias are much larger in EXPL longwave radiative balance cannot be better. Please correct this sentence

We now specify that RMSE and CRMSE improve and skipped the second part of the sentence.

-1138 "west- adn eastward ", citing only kelvin waves here is misleading as this implies eastward propagation, while African easterly waves are Rossby-type waves

We now mention Rossby and Kelvin waves.

-1175: remove "entrainment" as entrainment does not change environment directly, only through compensating subsidence

Done.

-1191: "improvement of moist bias through enhanced precip efficiency". This maybe, but is not certain at all, as the large-scale circulation (moisture advection changes) as you said yourself - btw this might also contribute why EXPL gets worse in the southern part

We added "likely" to the sentence in order to make it clear that this is one possible explanation.

-1204: "should"->"might" (note that global NWP/data assimilation does generally not apply a nesting strategy)

We rephrase this to: ... services are encouraged to evaluate...

Gregor Pante and Peter Knippertz

Reviewer #1 (Remarks to the Author):

Third Review of "Can resolving Sahelian thunderstorms improve global weather forecasts?" by Pante & Knippertz

The authors addressed my comments and suggestions very well, and I believe the current version of the manuscript is well suited for publication in Nature Communications. I do not need to see the manuscript again before its publication, as at this stage I have only a few minor comments listed hereafter.

l 62 : you could remove "largely" as only from these two figures, it is not so obvious which one of EXPLC 13 km and EXPLC 6.5 km is better (as I mentioned before I think this is an interesting point, although more investigation would be needed to address it fully, placing it beyond the scope of this study)

l 217-218 : no clear signal in the tropics; replace by "over some remote regions in the extra-tropics, including ..."

double check figure referencing, for instance l 92 replace S4 by "Supplementary Fig. 4"

Laurent Labbouz

Third review of the manuscript “Can resolving Sahelian thunderstorms improve global weather forecasts?” submitted to *Nature Communications*.

REVIEWER 1 (Laurent Labbouz)

The authors addressed my comments and suggestions very well, and I believe the current version of the manuscript is well suited for publication in *Nature Communications*. I do not need to see the manuscript again before its publication, as at this stage I have only a few minor comments listed hereafter.

l 62 : you could remove “largely” as only from these two figures, it is not so obvious which one of EXPLC 13 km and EXPLC 6.5 km is better (as I mentioned before I think this is an interesting point, although more investigation would be needed to address it fully, placing it beyond the scope of this study)

Done.

l 217-218 : no clear signal in the tropics; replace by “over some remote regions in the extra-tropics, including ...”

This sentence refers to the impacts, not the improvements of the two-way nesting. The impacts are strong in the tropics as well and we therefore prefer to stay with our wording here.

double check figure referencing, for instance l 92 replace S4 by “Supplementary Fig. 4”

Done.